# Entropy-Based Node Importance Identification Method for Public Transportation Infrastructure Coupled Networks: A Case Study of Chengdu

**DOI:** 10.3390/e26020159

**Published:** 2024-02-11

**Authors:** Ziqiang Zeng, Yupeng Sun, Xinru Zhang

**Affiliations:** 1Business School, Sichuan University, Chengdu 610065, China; 2022225025163@stu.scu.edu.cn; 2School of Management, Zhengzhou University, Zhengzhou 450001, China; zhangxr@gs.zzu.edu.cn

**Keywords:** coupled network, resilience, entropy, importance identification, public transportation infrastructure

## Abstract

Public transportation infrastructure is a typical, complex, coupled network that is usually composed of connected bus lines and subway networks. This study proposes an entropy-based node importance identification method for this type of coupled network that is helpful for the integrated planning of urban public transport and traffic flows, as well as enhancing network information dissemination and maintaining network resilience. The proposed method develops a systematic entropy-based metric based on five centrality metrics, namely the degree centrality (DC), betweenness centrality (BC), closeness centrality (CC), eigenvector centrality (EC), and clustering coefficient (CCO). It then identifies the most important nodes in the coupled networks by considering the information entropy of the nodes and their neighboring ones. To evaluate the performance of the proposed method, a bus–subway coupled network in Chengdu, containing 10,652 nodes and 15,476 edges, is employed as a case study. Four network resilience assessment metrics, namely the maximum connectivity coefficient (MCC), network efficiency (NE), susceptibility (S), and natural connectivity (NC), were used to conduct group experiments. The experimental results demonstrate the following: (1) the multi-functional fitting analysis improves the analytical accuracy by 30% as compared to fitting with power law functions only; (2) for both CC and CCO, the improved metric’s performance in important node identification is greatly improved, and it demonstrates good network resilience.

## 1. Introduction

Complex networks serve as important representations of many systems in the real world, including social networks [1], transportation networks [2,3], and power grids [4]. The Internet of Things (IoT) provides a sufficiently effective tool to build intelligent transport systems [5]. Sensors are an important part of IoT technology as they can collect and transmit a variety of data, including traffic flow, vehicle location and speed, and air quality data [6]. These data can be connected to public transport infrastructure systems through networks to support the operation and improvement of public transport services.

Meanwhile, the topology of a complex network is often highly nonlinear, highly interconnected, and scale-invariant [7,8]. Therefore, the identification of node importance in complex networks holds significant practical and research value. In recent years, node importance identification has been useful in many applications across various domains. It allows us to identify the most influential users in social networks, to enable the better monitoring of public opinions and the forecasting of major events [9]; to locate crucial intersections or nodes in transportation networks, to guide the optimization of bus network services and improve the capacity of the public transport system [10]; and, in the realm of power networks, to prioritize the monitoring and maintenance of critical nodes, thereby ensuring the reliability and stability of the power grid [11]. Additionally, it is possible to explore the functionality and disease mechanisms of biological systems by recognizing genes or proteins that play pivotal roles in processes such as protein interaction networks and gene-regulatory networks [12,13]. Assessing significant nodes in real networks is challenging due to the frequent coupling of different networks, interacting with each other and impacting the current network [14].

There have been many related studies on the issue of identifying node importance in single-layer networks. The concept of a structural hole refers to the gap between a group of indirectly connected nodes, which serve as intermediaries for information exchange and fill these structural holes. Based on the characteristics of structural holes, a heuristic algorithm based on local topological information was created for the identification of node importance in the symmetric adjacency matrices of undirected and unweighted networks [15]. Moreover, by improving the structural holes, Liu et al., proposed an algorithm for the identification of important nodes in complex networks that impact security by taking into account the global network structure and nodes’ structural hole features [16]. Wan et al., utilized multiple relationships and global network topology features in complex networks and classified heterogeneous nodes into core and auxiliary layers based on the node type. They quantified the importance of the auxiliary layer by determining the weighted interlayer influence based on different levels of connectivity, thereby measuring the importance of core layer nodes [17]. However, structural holes mainly analyze nodes from the perspective of local information.

In all of the above studies, the importance of each independent node was assessed on the basis of the network topology. However, node importance is not only influenced by the attributes of the nodes themselves, but also by the attributes of their neighboring nodes [18]. Zhong et al., proposed the Local Degree Dimensionality (LDD) method, which calculates the node importance based on the number of neighboring nodes in each layer of the central node. This method creatively combines the number of neighboring nodes in each layer and the incremental and decremental rates to derive the value of a node’s local degree dimension [19]. Shannon entropy (also known as information entropy) is a concept in information theory that is used to measure the uncertainty or amount of information in a random variable [20]. Based on this, Yu et al., developed a new measure called the node spreading entropy, which considers the clustering coefficients of nodes and the influence of both first-order and second-order neighboring nodes on node importance. From an entropy perspective, this approach identifies important nodes [21]. In addition, Sun et al., selected four indices reflecting different characteristics of the network structure based on complex network theory. They then calculated the weights of these indices using information entropy theory and then obtained the node importance through a weighted average method. This approach provides an adaptive node importance identification method based on entropy [22]. By introducing entropy, the above studies considered both the overall and local information of the network in the node importance identification task, which rendered the evaluation index more accurate and objective. However, all such identification tasks are performed based on single-layer networks, without taking into account the real-life coupling between networks; there is also a lack of research on public transportation networks.

The node importance identification task in coupled networks aims to identify and measure the importance of nodes in multiple interconnected and interdependent networks. However, there are relatively few studies available on node importance identification in coupled networks. This is because coupled networks have unique characteristics that result in complex dynamic behaviors and nonlinear interactions, making the node importance identification task more challenging. In real-world scenarios, many network systems consist of multiple coupled networks, such as the coupling between the power generation network and the information network in the electric power system [23], and the coupling between different modes of transportation in the transport network [24]. Identifying the importance of nodes in these networks requires the consideration of the interactions and coupling effects among the networks. To address the challenges of identifying node importance in coupled networks, researchers have explored new methods and metrics. For instance, the Enhanced Similarity Index (ESI) is proposed to measure inter-layer coupling relationships by considering the nodes’ own neighbors and the common neighbors of the nodes in two temporal layers. An attenuation coefficient is introduced to accurately describe the inter-layer coupling relationships. Based on this, an Attenuation-Based Super-Adjacency Matrix (ASAM) temporal network modeling method based on the attenuation of inter-layer coupling strengths has been used to identify the importance of nodes in the temporal network, by calculating the centrality of the eigenvectors of nodes in each temporal layer of the temporal network [25]. Liu et al., explored overlapping clusters in power grids based on a link-based segmentation method and identified the node importance in a single-layer network by using the Betweenness Centrality Based on Neighbor Nodes (BCBNN) algorithm. Then, they proposed a node importance identification method that considered the single-layer network topology, grid overlap structure, and dual-grid coupling [26]. Some results have been achieved by assessing the importance of nodes in coupled networks, but entropy-based metrics are still missing when considering the global information of nodes. A comparison of related studies is shown in Table 1.

The major contributions of this study can be delineated as follows.

Based on the modernized multi-level and multi-modal public transportation infrastructure network in Chengdu, a bus–metro coupled network containing 10,652 nodes and 15,476 edges is constructed and a network dataset is formed.The analysis of the coupled network is carried out by fitting the model with multiple functions, which improves the analytical accuracy by 30% compared to fitting with power-law functions only.This study introduces the entropy-based node importance identification and assessment system. Five centrality metrics are improved based on entropy for node importance identification in coupled networks. Moreover, this research takes the Chengdu bus and subway coupled network as a case study, and four network resilience metrics are selected—the maximum connectivity coefficient, network efficiency, susceptibility, and natural connectivity [27]—to evaluate the performance of the system.

The rest of this paper is organized as follows: Section 2 is the problem statement; Section 3 defines the network resilience measures, traditional centrality metrics, and entropy-based node importance identification methods; Section 4 constructs and analyzes the public transportation infrastructure coupled network in Chengdu; Section 5 identifies nodes’ importance through different metrics and compares them; finally, relevant conclusions and future research are given in Section 6.

## 2. Problem Statement

A large number of node importance identification methods have been proposed in previous studies, but few studies have been conducted for public transportation infrastructure coupled networks. By identifying important nodes in public transportation infrastructure coupled networks, resources and investments can be allocated rationally and efficiently in order to improve the operational efficiency of the public transportation system. For highly coupled networks, the condition and operation of important nodes can have a significant impact on the whole system, so prioritizing the maintenance and improvement of these nodes is key.

In this study, entropy is used to improve the traditional centrality metrics, and the entropy-based improvement simultaneously considers the importance of the nodes themselves and the influence of neighboring nodes on them. Then, a total of 10 metrics are used to identify the importance of the network nodes, and experiments are carried out on the case of Chengdu’s bus–subway network (abbreviation: CBSN) to verify the effect of the entropy-based improvement on the performance of the metrics by using the four network resilience assessment metrics.

The workflow diagram of this study is shown in Figure 1.

## 3. Related Work

### 3.1. Network Description

In this study, the complex network as a whole is defined as G=(V,E), the set of nodes in the network is $V=\{v_{1},v_{2},\ldots ,v_{n}\}$, and the set of edges is $E=\{e_{ij}|e_{ij}=f\left(v_{i},v_{j}\right),i,j=1,2,3,\ldots ,n\}$. $e_{ij}$ denotes the connectivity between nodes $v_{i}$ and $v_{j}$. The relevant symbols can be found in Appendix A, and the same applies to the other symbols below, and no further explanations will be made thereafter.
(1)$$e_{ij}=\left\{\begin{array}{c} 1,\mathrm{There}\;\mathrm{exists}\;\mathrm{an}\;\mathrm{edge}\;\mathrm{between}\;\mathrm{nodes}\;v_{i}\;\mathrm{and}\;v_{j}\\ 0,\mathrm{There}\;\mathrm{is}\;\mathrm{no}\;\mathrm{edge}\;\mathrm{between}\;\mathrm{nodes}\;v_{i}\;\mathrm{and}\;v_{j} \end{array}\right.$$

In this study, the Space L method is used to construct the coupling network [28], with the stations in the lines as nodes. When the stations $v_{i}$ and $v_{j}$ are neighboring nodes in the same line, there exists a connecting edge between the two points, i.e., $e_{ij}=1$. In order to abstract the real network, this study makes the following assumptions.

In the CBSN, all edges are undirected because the vast majority of routes exist both upward and downward, so the beginning and end of a route can be equated.In this study, we only consider the topology of the coupled network and do not consider the influence of the distance, train frequency, and passenger flow in the actual lines, so the network is an unweighted network and all edges are of length 1.

### 3.2. Measuring Network Resilience

Network resilience is the ability of a network to remain robust and adaptive in the face of varying loads, failures, attacks, or changes. It relates to aspects of network flexibility, transmission efficiency, and vulnerability. The improvement of the network resilience can ensure that the network has high availability, reliability, and stability and can effectively respond to unexpected situations and high load pressure, thus ensuring excellent performance and a complete topology structure in the network [29]. Therefore, it is important to accurately assess the resilience of the network. In this study, four metrics—the maximum connectivity factor, network efficiency, susceptibility, and natural connectivity—are used to assess the network resilience.

#### 3.2.1. Maximum Connectivity Coefficient

In complex networks, the maximum connectivity coefficient is defined by the ratio of the number of nodes in the maximum connectivity subgraph of the network to the total number of nodes. It can be used to measure the overall connectivity and stability of the network: the larger the maximum connectivity coefficient, the better the overall connectivity of the network. Moreover, the maximum connectivity coefficient reflects the most important nodes or groups in the network as they have the greatest impact on the overall connectivity [30].
(2)$$MCC=\frac{R}{N}$$
where *N* is the number of nodes in the initial state of the network and *R* is the number of nodes in the maximum connectivity subgraph of the remaining network after an attack. The interval of MCC is [0, 1]. When MCC is equal to 1, every node in the network is directly or indirectly connected to other nodes, forming a globally connected network. Meanwhile, when MCC is close to 0, there is no connection between the nodes in the network, and the network presents a decentralized and disconnected structure.

#### 3.2.2. Network Efficiency

In complex networks, network efficiency is a measure of the rapidity of network transmission and information dissemination. The improvement of network efficiency is important in speeding up information dissemination and improving network communication, as well as increasing the robustness and reliability of the system. Therefore, in complex networks, improving the network efficiency is often an important goal in network design and optimization. It is commonly used to assess the information exchange efficiency and resource utilization efficiency in a network. Global efficiency is a commonly used network efficiency metric [31]. It measures the speed of information transfer between all pairs of nodes in a network, i.e., the efficiency of global information transfer. The calculation of the global efficiency is usually based on the shortest path length between node pairs.
(3)$$\eta =\frac{1}{N(N-1)}\sum \frac{1}{L_{ij}}$$
where $L_{ij}$ is the shortest distance between pairs of connected nodes $v_{i}$ and $v_{j}$ in the network, and *N* is the total number of nodes in the network. In this study, in order to compare the identification effect of different metrics under different network resilience assessment methods, the network efficiency is calculated in the form of a ratio as follows:(4)$$NE=\frac{\eta }{\eta_{0}}$$
where $\eta_{0}$ is the global network efficiency in the initial state of the network and η is the global network efficiency of the network after an attack. The closer the value of NE is to 1, the smaller the efficiency loss after the network suffers an attack, and the closer the value of NE is to 0, the larger the efficiency loss.

#### 3.2.3. Susceptibility

The susceptibility (or vulnerability) of a network is the ease with which nodes or edges in the network can be damaged or removed, thus causing the network to become simpler or the connectivity and efficiency to deteriorate after an attack. When there are many vulnerable nodes or edges in a network, the destruction of these nodes or edges may result in a disruption or reduction in the network’s connectivity, thereby reducing or even eliminating the maximum subset of the network’s connectivity. This may lead to the degradation of the network’s functionality and interruptions or delays in information delivery, and it may even prevent the entire network from functioning properly [32].
(5)$$S=\sum_{s<R}\frac{s^{2}n_{s}}{N}$$
where *s* is the number of nodes in the subgraph of the network, $n_{s}$ is the number of subgraphs with subgraph nodes of size *s*, and *R* is the number of nodes in the maximum connected subgraph of the network. A network’s susceptibility tends to increase and then decrease with the increase in removed nodes, which is due to the fact that when a small number of nodes are removed, the network may experience the removal of some key nodes that play an important role in the network connectivity and information transfer. Therefore, the removal of these critical nodes may lead to the breakage or loss of the functionality of the network. Thus, as more nodes are removed, the vulnerability increases first. However, as more nodes are deleted, the structure and functionality of the network stabilizes again, as the critical nodes are removed and the remaining nodes are not sufficiently important; therefore, their vulnerability decreases again.

#### 3.2.4. Natural Connectivity

Regarding the internal structural properties of complex networks, natural connectivity reflects the redundancy of alternative pathways in the network by calculating the weighted sum of the number of closed loops of different lengths in the network, which is mathematically represented as a particular form of average eigenroot. This can be directly derived from the eigenspectrum of the network adjacency matrix, and it thus has clear physical significance and a concise mathematical form [33].
(6)$$\lambda =\ln\frac{1}{N}\sum_{i=1}^{N}e^{\lambda_{i}}$$
where λ represents the natural connectivity, *N* represents the number of nodes in the network, and $\lambda_{i}$ represents the characteristic root corresponding to the *i*th node in the network adjacency matrix. This study aims to compare the evaluation effectiveness of different metrics under various network assessment methods. Hence, the natural connectivity is computed in a proportional form as follows:(7)$$NC=\frac{\bar{\lambda }}{{\bar{\lambda }}_{0}}$$
where ${\bar{\lambda }}_{0}$ denotes the initial natural connectivity of the network, and $\bar{\lambda }$ represents the natural connectivity of the network after an attack.

### 3.3. Centrality Metrics

Import nodes are key to maintaining the stability and robustness of the network. Their removal or failure in the network may lead to the rupture of the network structure, resulting in the disruption of information transfer or degradation of the overall functionality of the network [34]. In order to identify the importance of nodes, this study defines important nodes as nodes that have a large impact on the resilience of the network. In previous studies, many methods have been proposed to identify the importance of nodes; in this study, five centrality metrics are selected for the effectiveness and network analysis, which are defined as follows.

#### 3.3.1. Degree Centrality (DC)

DC is calculated based on the degree of a node (i.e., the number of node connections). A higher value of degree centrality indicates that the node has more connections in the network and has a greater influence on information transfer and the overall functioning of the network [22].
(8)$$DC_{i}=\frac{\sum_{i\neq j}e_{ij}}{n-1}$$
where $DC_{i}$ is the degree centrality of node $v_{i}$, and $v_{j},v_{j}\in V$.

#### 3.3.2. Betweenness Centrality (BC)

BC identifies the importance of a node based on how often the node acts as a bridge or mediator in the network. It measures the number of times that a node is passed by the shortest path between other nodes in the network. Specifically, it calculates the ratio of the number of the shortest paths through a node to the number of all shortest paths, which in turn measures the degree to which the node acts as an intermediary [35].
(9)$$BC_{i}=\sum_{i\neq j\neq k}\frac{\sigma_{jk}\left(i\right)}{\sigma_{jk}}$$
where $BC_{i}$ is the betweenness centrality of node $v_{i}$, $\sigma_{jk}$ is the number of shortest paths from node $v_{j}$ to node $v_{k}$, and $\sigma_{jk}\left(i\right)$ is the number of shortest paths through node $v_{i}$. BC can denote the bridging role or intermediation of a node between different parts of a network in terms of information transfer and connectivity. Nodes with higher betweenness centrality usually play a key role in the overall function and influence of the network.

#### 3.3.3. Closeness Centrality (CC)

CC is calculated based on the average distance between a node and other nodes. In simple terms, closeness centrality measures the direct accessibility between a node and other nodes. Closeness centrality can be defined as the reciprocal of the average distance between a node and other nodes. The distance can be the distance of the shortest path or other defined distance metrics. CC measures the closeness of a node to other nodes in the network; the higher the value of CC, the shorter the distance between the node and other nodes and the more efficient the information transfer and influence propagation [21].
(10)$$CC_{i}=\frac{n-1}{\sum_{i\neq j}L_{ij}}$$
where $CC_{i}$ is the closeness centrality of node $v_{i}$.

#### 3.3.4. Eigenvector Centrality (EC)

EC calculates the importance of a node based on the strength of its connection to its neighboring nodes. In the calculation of eigenvector centrality, the importance of a node is related to the importance of its neighboring nodes [21]. The importance of a node in the network depends on how important the nodes connected to it are. If a node is connected to other important nodes, the eigenvector centrality of this node increases accordingly.
(11)$$EC_{i}=x_{i}=c\sum_{j\in V(i)}e_{ij}x_{j}$$
where V(i) refers to the set of neighboring nodes of node $v_{i}$, *c* refers to a constant, and $x_{i}$ and $x_{j}$ represent the importance of nodes $v_{i}$ and $v_{j}$, respectively. $x={\left[x_{1},x_{2},x_{3},\ldots ,x_{n}\right]}^{T}$, which can be written in the following matrix form when the steady state is reached after many iterations:(12)x=cAx
where *x* represents the eigenvector corresponding to the eigenvalue $c^{-1}$ of matrix *A*.

#### 3.3.5. Clustering Coefficient (CCO)

The CCO is a metric used to measure the degree of aggregation of nodes in a network, which indicates the connectivity between a node’s neighboring nodes, i.e., the probability that a node with degree *k* forms a connection among its neighboring nodes. The clustering coefficient can be used to measure the closeness of the local community or subgraph in which a node is located in the network [36]. It can be divided into the global clustering coefficient and local clustering coefficient.

Global clustering coefficient: The global clustering coefficient is the average of the clustering coefficients of all nodes in the network, reflecting the overall degree of aggregation of the network.Local clustering coefficient: The local clustering coefficient is the ratio of the number of actual connections to the number of possible connections between the neighboring nodes of node $v_{i}$, which is used to measure the degree of aggregation of a particular node.

The local clustering coefficient of node $v_{i}$ can be obtained by the following calculation:(13)$$CCO_{i}=\frac{2\times E_{i}}{N_{i}\left(N_{i}-1\right)}$$
where $E_{i}$ is the number of edges formed between the neighboring nodes of node $v_{i}$, and $N_{i}$ is its degree value, i.e., $N_{i}=\sum_{i\neq j}e_{ij}$, and $v_{j},v_{j}\in V$.

The interval of CCO is [0,1]. A CCO close to 1 indicates that the node’s neighboring nodes are closely connected to each other and a stronger community structure exists; a CCO close to 0 indicates that the neighboring nodes are more detached from each other and lack a tight community structure.

### 3.4. Entropy-Based Node Importance Identification Method

#### 3.4.1. Identification Process

Through the above node importance identification metrics, the nodes in the coupled network are assessed by their importance, and the set of importance identification results with different metrics is obtained, $P_{I}=\left[I\left(1\right),I\left(2\right),\ldots ,I\left(n\right)\right]$, where I=DC,BC,CC,ECorCCO denote the nodes using different metrics for importance identification.

#### 3.4.2. Data Normalization

Different metrics have different calculation methods and therefore have different scale ranges. They can be modified to have similar scales by using data normalization, which enables the data to be processed more effectively afterwards. At the same time, normalization removes unit differences between different features in the data, making the data more comparable. Therefore, before the entropy-based metric improvement, the data of different metrics need to be normalized as defined below:(14)$$I_{b}\left(i\right)=\frac{I\left(i\right)-min\left(P_{I}\right)}{max\left(P_{I}\right)-min\left(P_{I}\right)}$$
where I(i) denotes the importance obtained by evaluating node $v_{i}$ with metric *I*; $max\left(P_{I}\right),min\left(P_{I}\right)$ denote the maximum and minimum values obtained by evaluating all the nodes in the network with metric *I*, respectively; and $I_{b}\left(i\right)$ denotes the value after normalization.

#### 3.4.3. Modeling Method

Entropy has been introduced in node importance identification in previous studies [21,22]. By introducing the notion of information entropy, the global information of the network can be considered more comprehensively. Therefore, this study improves the five traditional centrality metrics based on entropy.
(15)$$I_{e}\left(i\right)=\frac{I_{b}\left(i\right)}{\sum_{j\in V}I_{b}\left(j\right)}$$
(16)$$I_{E}\left(i\right)=-\sum_{j\in V(i)}I_{e}\left(j\right)\ln I_{e}\left(j\right)$$
where *V* represents the set of all nodes, V(i) represents the set of neighboring nodes of node $v_{i}$, $I_{e}\left(i\right)$ represents the ratio of the importance of node $v_{i}$ to the sum of the importance of all the nodes under the assessment of metric *I*, and $I_{E}\left(i\right)$ represents the importance of node $v_{i}$ after the entropy-based improvement under the assessment of metric *I*.

### 3.5. An Example of an Entropy-Based Node Importance Identification Method

By calculating different metrics, we can obtain the ranking of the network nodes under different metrics, remove nodes in the network through the ranking, and measure the network’s resilience during the removal process. The greater the loss of network resilience under the same number of removals, the more accurate the assessment of the corresponding metrics. As shown in Figure 2, we take a small-scale network as an example for calculation.

See Appendix B for detailed calculations.

In the calculations, it can be seen that with the deletion of node 5, the overall resilience of the network changes. The maximum connectivity coefficient becomes 0.75, which means that the connectivity of the network decreases by 25%, and the network efficiency becomes 0.7 of the initial network, which means that the transmission efficiency of the network node decreases by 30%. Compared with the initial network, the susceptibility is improved from 0 to 0.14, which is logical; with the removal of node 5, the network becomes simpler and less susceptible to node attacks. Moreover, the natural connectivity of the network becomes 0.6 of the initial network, indicating that the robustness of the network decreases by 40%. Overall, the resilience of the network is greatly affected.

## 4. Network Analysis

### 4.1. Chengdu Bus–Subway Coupled Network Visualization

Chengdu is one of the largest cities in Southwest China and the capital city of Sichuan Province. With its large population and huge transportation needs, public transportation is of vital importance to this city. After years of construction and development, Chengdu’s public transportation system has been transformed into a modern multi-level and multi-modal transportation network including buses, subways, railways, and a BRT (Bus Rapid Transit System). The construction and operation of these transportation systems have provided Chengdu’s citizens with convenient and fast travel options [37]. As of August 2023, the Chengdu subway has opened a total of 16 lines and 356 subway stations, and the Chengdu bus system has opened 1407 lines and 10,841 bus stops, forming a huge public transportation network. Therefore, this study takes the CBSN as a case to construct a coupling network. Figure 3 shows the station map of the CBSN.

After obtaining the data sets of the currently running public transport operation lines and stations (including buses and subways) in Chengdu, the relevant data are cleaned, the night lines and special lines with a lower travel density are removed, and the lines with too few stations, as well as the self-looped lines, are removed. Then, according to the Space L method, the neighboring stations in the same line are set as connected edges. Finally, the CBSN, with 10,652 nodes and 15,476 edges, is constructed. The bus stops near the entrances and exits of the subway stations are merged with the adjacent subway stations to form coupling nodes, so as to achieve the coupling of the bus and subway networks.

### 4.2. Multifunction Fitting Analysis

In this study, we first calculate all the nodes in the network by using five metrics, DC, BC, CC, EC, and CCO, to obtain the distribution of node importance under different metrics. We then fit the distribution of node importance by using different functions; these include a uniform distribution (uniform), normal distribution (norm), t-distribution (t), log-normal distribution (lognorm), beta distribution (beta), exponential distribution (expon), Pareto distribution (pareto), generalized extreme value distribution (genextreme), double Weibull distribution (dweibull), gamma distribution (gamma), and log-gamma distribution (loggamma). Eleven functional distribution models are fitted and the results of the fits are compared using the Residual Sum of Squares (RSS) (see Figure 4). The RSS is the sum of the squares of the differences between the observed values and the predicted values of the regression model in a regression analysis. The RSS measures the degree of deviation between the fitted line and the actual observations; the smaller it is, the better the model fits the data. It is calculated as follows:(17)$$RSS=\sum_{i=1}^{N}{\left(y_{i}-f\left(v_{i}\right)\right)}^{2}$$
where $y_{i}$ represents the predicted value of node $v_{i}$ in the fitted function model, and $f(v_{i})$ denotes the corresponding actual value of the importance of node $v_{i}$ under the importance identification metric.

In Figure 4, it can be seen that the optimal models for different metrics are different, in which the optimal model for DC is the t-distribution model, the optimal model corresponding to EC and BC is the norm distribution model, and the optimal function model for the remaining two metrics is the beta distribution model. However, regardless of the metric, after multifunction fitting, the accuracy of the optimal function fit is improved compared to the accuracy of the node distributions obtained from the direct power-law function fitting in previous studies. Moreover, after calculating the RSS improvement value, it can be concluded that the average improvement is more than 30%. Then, the optimal fitting distribution function is used for the fitting of the different metrics, and the results are expressed in the form of a cumulative distribution function (CDF) for more intuitive results.

### 4.3. Fitting Result Discussion

In this study, we select five metrics DC,BC,CC,EC,CCO to identify the importance of nodes in the CBSN. After analyzing all the nodes in the network with different metrics (see Figure 5), we obtain the ranking of the nodes under the identification of these metrics (see Table 2, which only displays the node numbers of the ten highest-ranked nodes).

Based on the analysis of the metric distribution, the following conclusions can be drawn.

Most of the nodes in the network have low DC values and are connected to only a small number of other nodes. A small number of nodes have high DC values and are connected to a large number of other nodes. There are central nodes in the network with high degree values that may play an important role in the network. Due to the large number of nodes with low degree values, the network may be somewhat decentralized, i.e., there is no obvious core structure and it is a sparse network.The node betweenness centrality and eigenvector centrality distributions show extreme positive skewness, which means that the vast majority of nodes have very low values and only a very small number of nodes have high values, where approximately 80% of the nodes’ BC and EC values are close to 0. This distribution indicates the existence of a number of key nodes in the network, which play an important role in disseminating information, controlling flows, or connecting different communities. At the same time, most of the nodes have low values and may be relatively common nodes. Due to the large size of the network, there may be clusters of highly connected nodes or tight connections between some of the nodes.For closeness centrality, the results are more evenly distributed, implying a relatively balanced pattern of connectivity among the nodes in the network. There is no obvious central node or core cluster, and the whole network presents a more balanced structure. Such a network structure is common in some dispersed and decentralized systems.The clustering coefficient measures the degree of interconnectivity among the neighboring nodes of a node. In the CBSN, most nodes lack direct connections with each other, resulting in a clustering coefficient of 0. This also indicates that the network is highly unbalanced. Only a few nodes cluster together to form rich interconnections, while most nodes lack close neighbor connections.

## 5. Result Analysis

The experimental environment of this study is configured as follows: the operating system is Windows 11, the development language is Python 3.10.5, the CPU used is an Intel(R) Core(TM) i5-8250U CPU @ 1.60 GHz 1.80 GHz, and the memory used is 237 G. In the experiment, we assess and rank the importance of the nodes by using the five centrality metrics and their entropy-based improved metrics. We then select nodes for deletion sequentially from top to bottom based on the sorting results; due to the large number of nodes in the network, this study adopts proportional removal to carry out this task. A total of 11 subgroups (0%, 0.1%, 0.2%, 0.3%, 0.4%, 0.5%, 0.6%, 0.7%, 0.8%, 0.9%, 1%) are tested, i.e., the top n% of nodes are deleted based on the ranking of the metrics. Finally, the removed networks are assessed by using the four network resilience assessment metrics. The results are plotted as line graphs grouped according to the traditional metrics (see Figure 6) and the entropy-based improved metrics (see Figure 7). In the case of removing the same percentage, the the network resilience decreases more quickly, indicating that the removed nodes have a greater impact on the network, which is also more important.

### 5.1. Traditional Centrality Metrics

According to the results given in Figure 6a, we can obtain the following conclusions. Firstly, the effect of BC on the maximum connectivity coefficient is the largest in the range of removal percentages of 0 to 5%. This indicates that the intermediation between the nodes in the network plays a key role and positively affects the connectivity of the network. Further, after the removal ratio ratio exceeds 6%, we find that the effect of DC on the maximum connectivity coefficient becomes most prominent. This implies that the centrality values of the nodes in the network become a key determinant of the network connectivity.

The results in Figure 6b allow the following conclusions to be made. Firstly, as the percentage of removal increases, the network efficiency decreases by more than 90%. This indicates that the transmission function of the network is gradually and seriously affected during the process of node removal. In addition to this, we find that DC has the greatest degree of descent regardless of the removal ratio. This implies that the degree centrality is a more important assessment metric in maintaining network efficiency.

Based on the results given in Figure 6c, this study further discusses the relationship between susceptibility and different centrality metrics. Firstly, it can be observed that EC, CC, and CCO as assessment metrics do not accurately identify the importance levels of nodes to predict the susceptibility of the network. However, DC and BC can be used to identify important nodes that have a significant impact on susceptibility.

It is worth noting that although both BC and DC can have an effect on susceptibility, the effect of BC appears earlier. We observe an increase in susceptibility when a smaller number of nodes is removed, suggesting that the network has become more vulnerable. In contrast, the effect of DC on susceptibility is more pronounced, reaching a peak of 54 when the percentage of removal reaches 9%, implying that the network is already in its most vulnerable state.

According to the results given in Figure 6d, the influence of DC is the largest for the natural connectivity of the network, and the loss of natural connectivity is close to 50% when the removal ratio is 10%. The decrease is fast and then slow, which indicates that the more highly ranked nodes have the largest influence on the natural connectivity. The other four metrics have similar and much lower impacts on the natural connectivity than DC.

### 5.2. Improved Centrality Indicator

The same analysis is performed based on the improved metrics.

In Figure 7a, it can be seen that the impact of CCBE on the maximum connectivity coefficient of the network is the largest as the percentage of node removal increases, and the most important nodes obtained through the assessment of CCBE lose 45.7% of their maximum connectivity when the percentage of removal reaches 10%. Meanwhile, BCBE has only a slightly higher impact on the maximum connectivity coefficient than CCBE when the removal ratio is 3% and below.

From Figure 7b, firstly, it can be observed that the effect of CCBE on the network efficiency is the most significant at different removal ratios. When the removal ratio reaches 10%, only 13.1% of the network efficiency remains, which is the lowest value among all the metrics. This indicates that the clustering coefficient of the nodes has a very important effect on the network efficiency. In addition, the node importance as assessed by the metrics DCBE, BCBE, and CCOBE has a similar effect on the network efficiency. This implies that the degree centrality, median centrality, and clustering coefficient centrality of nodes play a similar role in the network efficiency. However, in contrast, ECBE has the smallest impact on the network efficiency; the network efficiency remains at 60.3% for the same percentage of node removal (10%), which is the highest value among these metrics.

This can be seen by analyzing Figure 7c. There is a significant difference in the effect of the different metrics on susceptibility, with DCBE and ECBE having almost no effect on network susceptibility at the current removal ratio, while the other three metrics all have a significant effect on susceptibility. For BCBE, the susceptibility peaks at 50.8 at a 6% removal ratio, while, for CCBE, the susceptibility rises as the percentage of nodes removed increases, and it reaches 52.8 at a 10% removal ratio, with the potential to continue to rise if nodes continue to be removed. Although CCOBE also has a significant effect on susceptibility, the effect does not appear until 7% and beyond, and it is weaker than that of CCBE and BCBE.

Based on Figure 7d, it can be concluded that the effects of CCBE,CCOBE and DCBE on the natural connectivity of the network are similar, and the rate of the decrease in the network connectivity under all three metrics is greatly improved when the node removal ratio is 1–2%, but this does not occur under the other removal ratios. This is due to the fact that, after the entropy-based improvement, the comprehensive identification performance of the different metrics improves, and the nodes or clusters that have a greater impact on the natural connectivity are all present in the range of 1–2%, and it may be the same among different metrics.

### 5.3. Comprehensive Comparison

From the above analysis, it is found that the impacts of the different network resilience assessment metrics are not the same. In order to horizontally compare the traditional centrality metrics with the entropy-based improved centrality metrics, this study selects the two node importance identification metrics that have the greatest impact on the network resilience for comparison (see Table 3). The results of the experiments according to the four network resilience assessment metrics are plotted as a line graph (see Figure 8), and we conduct a comprehensive comparison analysis.

Based on the analyzed results in Figure 8a, it can be concluded that different metrics affect the maximum connectivity coefficient to different degrees at different stages. The impact of BC is the largest at a node removal percentage of 0–4%. This means that the coupling network will lose approximately 20% of the nodes in this phase; these nodes play the roles of important bridges or key positions in the network. At a node removal ratio in the range of 5–6%, CCBE becomes the most influential metric on the maximum connectivity coefficient. CCBE reflects the degree of clustering of nodes and their importance to the connectivity in the network. At this point, the clustering characteristics of the nodes play a key role in the connectivity of the network. However, after the removal ratio exceeds 7%, the influence of DC suddenly increases and becomes the highest among all metrics. The degree centrality metric reflects the number of connections of a node in the network. This means that as more nodes are removed, nodes that are directly connected to other nodes have a greater influence on the network connectivity.

As seen in Figure 8b, the impact of DC and CCBE on the network efficiency is comparable at node removal ratios less than 5%, with CCBE having a slightly higher impact. In other words, both DC and CCBE contribute to the global efficiency of the network to some extent at this stage. The network efficiency may be jointly affected by both metrics within this range of node removal ratios. However, after the node removal ratio exceeds 6%, the impact of DC on the network efficiency becomes greater. This means that as more nodes are removed, DC becomes a key factor affecting the network efficiency. On the other hand, at the current removal ratio, BC and DCBE have a similar impact on the network efficiency, but it is smaller than the influence of DC and CCBE. This implies that, at this stage, BC and DCBE contribute relatively little to the network efficiency.

As shown in Figure 8c, all four metrics have some degree of influence on the susceptibility of the network. At different node removal ratios, each metric exhibits different characteristics. First, it can be observed that BC is the metric with the fastest impact on network susceptibility. This means that when a small number of nodes are removed, BC affects the network susceptibility rapidly. Nodes with high betweenness centrality play the roles of bridges or key connections in the network, and thus their removal causes information dissemination in the network to become more difficult. Second, at the current node removal ratio, although DC has the greatest degree of impact, its impact rapidly increases only after the node removal ratio exceeds 5%. Nodes with higher degree centrality connect to more nodes, and the impact of degree centrality is only gradually apparent as more nodes are removed. Then, as the proportion of node removals increases, the effect of BCBE on susceptibility increases, and the network’s susceptibility decreases rapidly, indicating that the stability of the network structure is decreasing rapidly. Finally, regarding CCBE, although its impact is not the greatest, it rapidly increases when a small number of nodes are removed, while having a similar degree of impact as DC.

As shown in Figure 8d, the influence of DC on natural connectivity is the largest under different node removal ratios, indicating that, for natural connectivity, the node’s degree in the network is the main factor that affects it. Specifically, nodes with higher degree centrality will have a greater impact on natural connectivity, while, for the other metrics, the entropy-based improvement to take into account the attributes of neighboring nodes has a positive influence on the metrics, resulting in an increase in their influence on the natural connectivity of the network. Meanwhile, it can be observed that in all the curves, the decrease in natural connectivity shows an initially fast and then slow pattern; it can be concluded that, among the different metrics, the highest ranking nodes have a greater influence, i.e., a small number of nodes are damaged, which will lead to a rapid decrease in natural connectivity.

According to the above analysis of the performance of the different node importance identification metrics under different network resilience metrics and in different attack phases, we select nodes that have a greater impact on the network resilience under different resilience assessment metrics as the most important nodes (see Figure 9).

### 5.4. Effectiveness Comparison

Based on the analysis above, in order to provide a more intuitive evaluation of the effectiveness of the entropy-based improvements, this study compares the traditional centrality metrics with their corresponding entropy-based improved metrics. By assessing the network resilience using different metrics, the experimental results are compared, as shown in Figure 10.

Based on the comparison results, it can be seen that after the entropy-based improvement of the centrality metrics, there is a significant improvement in some of the metrics.

For DC, its performance decreases after the improvement, which is due to the fact that DC can visually represent the degree of association of a node with other nodes. Moreover, for network resilience, the degree centrality of a node itself is much more influential than that of its neighbors; even if it is improved by entropy, it will not be able to improve its evaluation—rather, it leads to a decrease.For CC and CCO, the performance of the metrics is greatly enhanced by the improvement, which indicates that, in the coupled network, the CC and CCO of the nodes have less influence on the network resilience, so, after the entropy-based improvement, the importance of the node and its neighboring nodes is more comprehensively taken into account in CCBE and COBE, which improves the performance of the identification of these two metrics.For BC and EC, after the improvement, the identification of the importance of network resilience is essentially unchanged, which may be due to the fact that the two metrics have already taken into account the information of neighboring nodes in the calculation itself; thus, even after the improvement, no significant change can be obtained.

According to the above analysis, we can make the following suggestions for the public transportation infrastructure coupled network in Chengdu.

The fitting of the distribution of important nodes can help the government to evaluate and plan the network more accurately. Based on the characteristics of the distribution of important nodes, the government can carry out reasonable road network planning and station layout and route optimization, and it can achieve lower maintenance costs while making the public transportation network more efficient and convenient.Important nodes play a key role in the public transportation network, so the maintenance and management of these nodes should be strengthened, and different importance identification metrics should be used to identify important nodes at different stages of the network under attack. Meanwhile, the government can provide additional resources and support to ensure the normal operation and good service quality of important nodes.

## 6. Conclusions and Future Research

This study constructs an undirected and unweighted bus–subway coupling network in Chengdu, analyzes the network by using five traditional centrality indicators, and fits the network with different functions. It then evaluates the effectiveness of the function model fitting by using the RSS and selects the optimal fitting model to analyze the network. Then, this study improves the traditional centrality indexes based on entropy, obtains five improvement metrics, assesses their effectiveness by using four network resilience evaluation metrics in different metric groups, and selects the optimal two metrics for a comparative analysis among the traditional metrics and the improvement metrics. The final results are as follows. (1) For the traditional centrality metrics, the performance of DC decreases slightly with the entropy-based improvement, but the performance of CC and CCO is substantially improved, indicating that, for metrics that fail to take into account the information of neighboring nodes, their importance identification performance can be mostly improved through entropy-based improvement. (2) Among the traditional metrics, DC has the best performance and the greatest impact on network resilience, and, among the metrics improved based on entropy, CCBE performs the best and has the greatest impact on network resilience. (3) The metrics that have the greatest impact on network resilience are different at different stages of node removal, and the entropy-based improvement metric performs well for network susceptibility; meanwhile, for the maximum network connectivity coefficient, network efficiency, and natural connectivity, it only outperforms the traditional centrality metrics in some stages.

There are some limitations in this study that need to be further investigated in the future. First, regarding the constructed network, this study only considers the topology of the network, and it ignores the distance, the passenger flow, and the effect of POI (point of interest) on the nodes in the actual network, so the connection with the actual network needs to be further strengthened. Secondly, in the actual network, for some lines, the upward and downward line stations are not the same, and there is a gap in the passenger flows of different lines, so the actual network is more likely to be a directed weighted network, which can be further analyzed in future research.

## Figures and Tables

**Figure 1 entropy-26-00159-f001:**
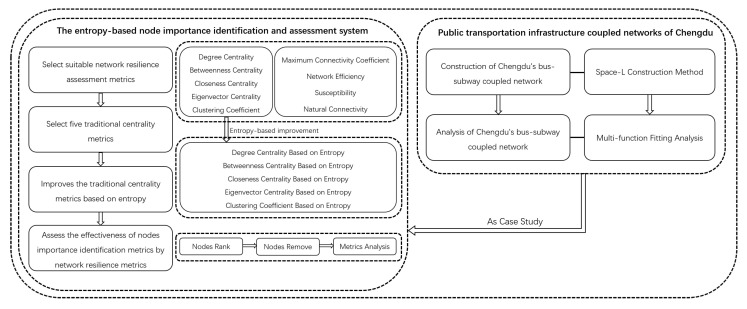
The workflow diagram of this study.

**Figure 2 entropy-26-00159-f002:**
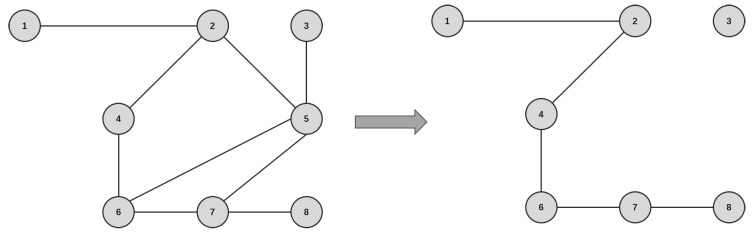
An example small-scale network. The numbers in the picture represent different nodes, which are connected directly by the edges.

**Figure 3 entropy-26-00159-f003:**
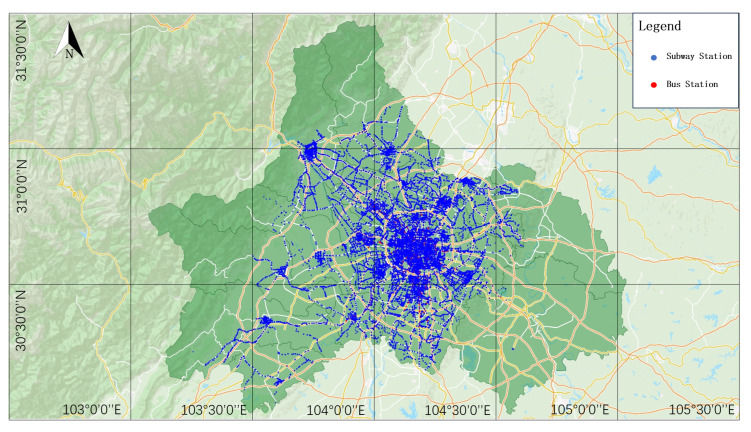
Chengdu bus and subway coupling network site map.

**Figure 4 entropy-26-00159-f004:**
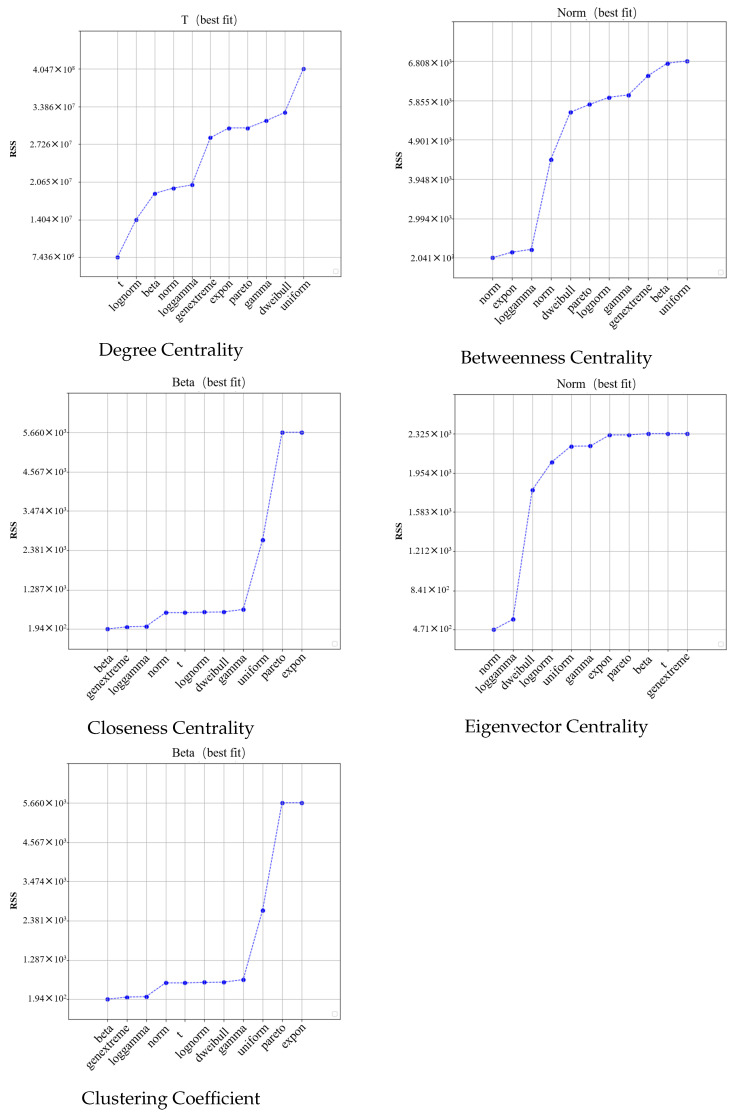
RSS for fitting of different function models based on different indicators.

**Figure 5 entropy-26-00159-f005:**
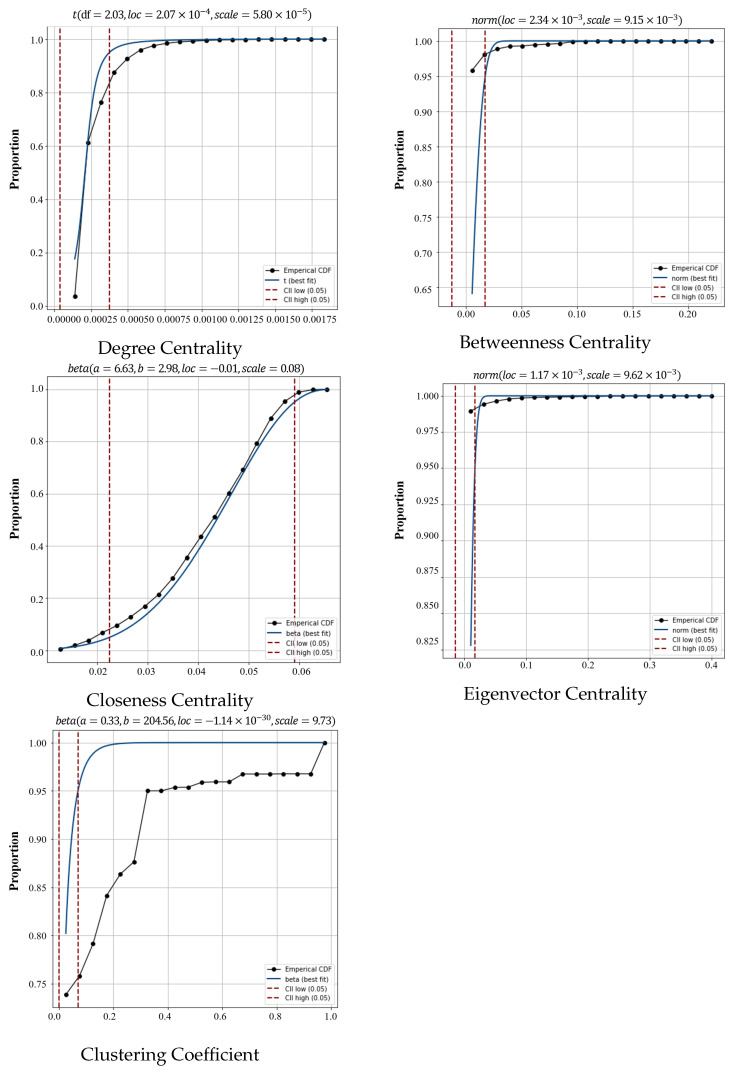
Fitting results of the selected optimal distribution under different metrics. The results are presented in the form of CDFs (i.e., $F_{X}\left(x\right)=P\left(X\leq x\right)$), with CII low (0.05) denoting the lower line of the 95% confidence interval and CII high (0.05) denoting the upper line.

**Figure 6 entropy-26-00159-f006:**
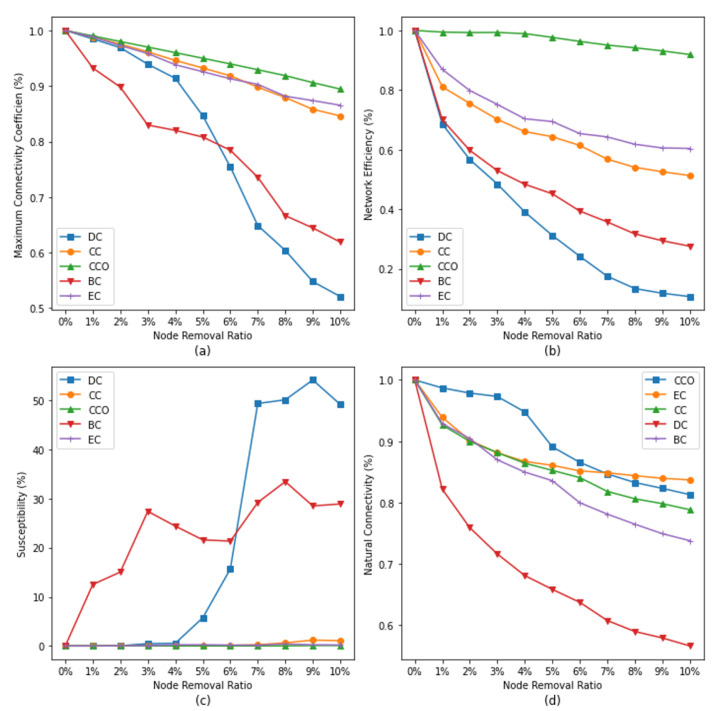
Impact of removing nodes on network resilience based on centrality metrics. Subgraphs (**a**–**d**) represent the changes of different network resilience indicators with the increase of node removal ratio under the evaluation of traditional centrality indicators, with subgraph (**a**) representing the Maximum Connectivity Coefficient, subgraph (**b**) representing Network Efficiency, subgraph (**c**) representing Susceptibility, and subgraph (**d**) representing Natural Connectivity.

**Figure 7 entropy-26-00159-f007:**
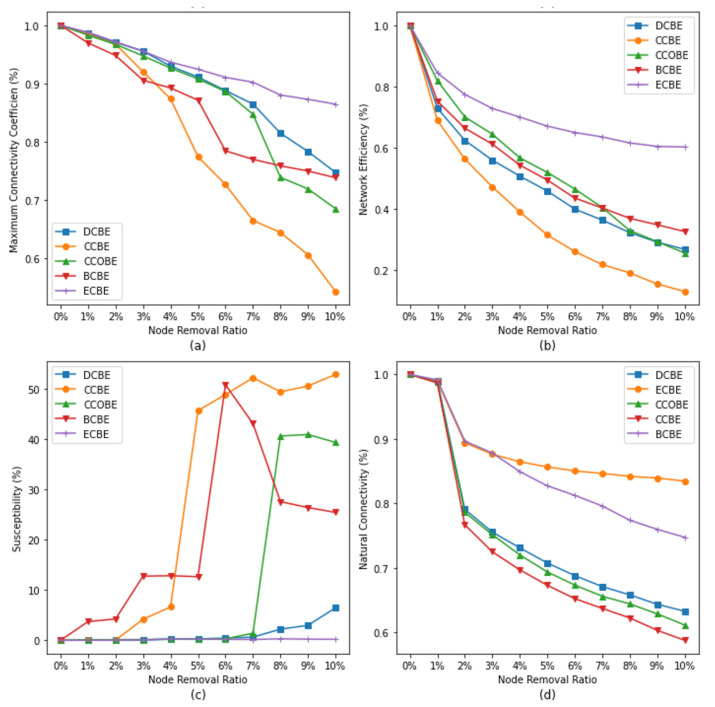
Impact of removing nodes on network resilience based on entropy-based improved metrics. Subgraphs (**a**–**d**) represent the changes of different network resilience indicators with the increase of node removal ratio under the evaluation of entropy-based improvement indicators, with subgraph (**a**) representing the Maximum Connectivity Coefficient, subgraph (**b**) representing Network Efficiency, subgraph (**c**) representing Susceptibility, and subgraph (**d**) representing Natural Connectivity.

**Figure 8 entropy-26-00159-f008:**
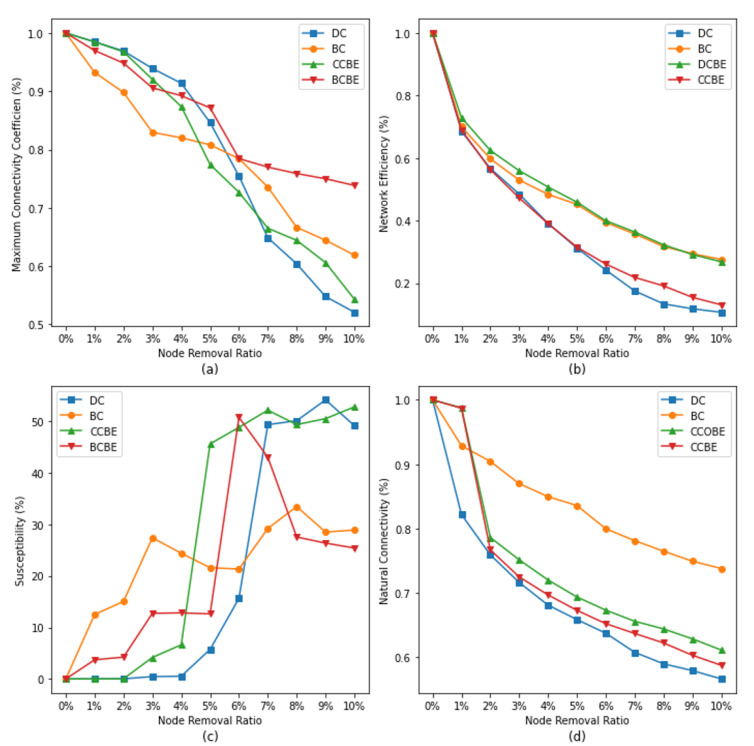
A comprehensive comparison between traditional centrality indicators and improved entropy-based indicators. Subgraphs (**a**–**d**) represent the changes of different network resilience indicators with the increase of node removal ratio under the four best indicators of centrality and entropy-based improvement indicators, with subgraph (**a**) representing the Maximum Connectivity Coefficient, subgraph (**b**) representing Network Efficiency, subgraph (**c**) representing Susceptibility, and subgraph (**d**) representing Natural Connectivity.

**Figure 9 entropy-26-00159-f009:**
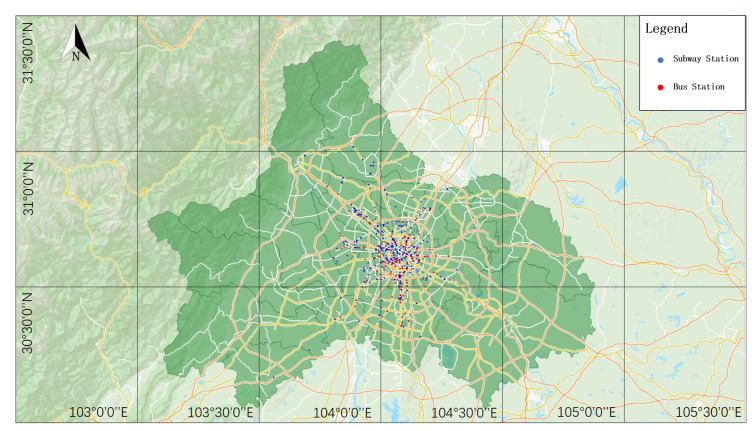
Important nodes in the CBSN.

**Figure 10 entropy-26-00159-f010:**
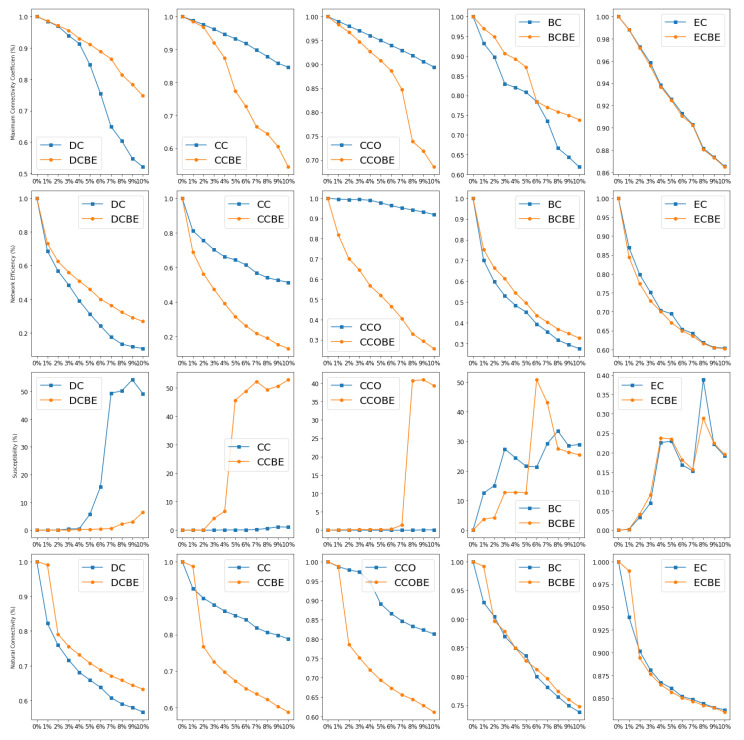
Effectiveness comparison.

**Table 1 entropy-26-00159-t001:** Comparison of domestic and international related studies.

Related Study	Node Local Information	Node Global Information	Based on Entropy	Network Type	Modeling Method
Zhong (2022) [19]	√	√		Single-layer	SI
Yu (2022) [21]	√	√	√	Single-layer	SIR and SIRS
Sun (2020) [22]	√		√	Single-layer	Attack simulation
Jiang (2022) [25]		√		Multi-layer	ASAM
Qi (2021) [27]	√			Multi-layer	Hybrid traffic network model
Yang (2020) [15]	√			Single-layer	Configuration model
This Study	√	√	√	Multi-layer	Multifunction fitting model and attack simulation

**Table 2 entropy-26-00159-t002:** Node numbers of the top ten nodes ranked according to different metrics.

Rank	DC	BC	CC	EC	CCO
1	22	644	649	78	17
2	78	624	78	806	26
3	272	806	644	644	29
4	384	366	806	805	60
5	543	8733	336	643	170
6	729	643	643	1411	181
7	806	5127	1411	4218	200
8	336	3196	77	77	222
9	27	4507	10,580	336	229
10	226	543	478	649	257

The network is generated by sequentially assigning a unique serial number to each node, and the nodes are denoted by their serial numbers in the table.

**Table 3 entropy-26-00159-t003:** Comprehensive comparison metrics.

Experimental Group	Maximum Connectivity Coefficient	Network Efficiency	Susceptibility	Natural Connectivity
Traditional Centrality				
Metrics	DC, BC	DC, BC	DC, BC	DC, BC
Improved Centrality				
Metrics	BCBE, CCBE	CCBE, DCBE	BCBE, CCBE	CCBE, CCOBE

## Data Availability

The data used in this study were collected and collated by the author, who collected and analyzed the data of the Chengdu bus and metro network through Python 3.10.5.

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
