# Peer review of "Entropy-Based Node Importance Identification Method for Public Transportation Infrastructure Coupled Networks: A Case Study of Chengdu"

_entropy, 2024, doi:10.3390/e26020159_

Round 1
Reviewer 1 Report
Comments and Suggestions for Authors
Thank you for your interesting work. I would like to suggest a few revisions as follows.
For Sections 3.2, 3.3, and 3.4, if a small toy network can be used to demonstrate how these metrics are calculated, it will be very helpful for readers to better understand the numerical case study. It can be done as an additional Section 3.5.
Letter sizes in Figure 3 seem too small so that it can be read clearly. Please revise. Same for Figure 4.
In Figure 7c, BCBE’s susceptibility should be discussed in the manuscript.
When the proportions of nodes are removed, the manuscript does not have discussions regarding how the node selections are made.
Author Response
Dear Reviewer,
Thanks very much for taking your time to review this manuscript. We really appreciate all your comments and suggestions. Those comments are all valuable and very helpful for revising and improving our manuscript. All the comments have been carefully considered and we have made our best efforts to revise the whole manuscript. Please find our detailed response to your comments as below. We appreciate all the comments and suggestions and hope that our revisions have met all the requirements. Please feel free to inform us if there are any problems in our revised manuscript.
Best regards,
Sincerely yours
Ziqiang Zeng, Yupeng Sun, Xinru Zhang.

Reviewer 2 Report
Comments and Suggestions for Authors
Dear Authors,
I read the content of the manuscript with great interest. Identification of the importance of public transport nodes significantly influences the process of making decisions regarding investing in the equipment of these nodes to increase (or at least maintain) their role in the transport system. The topic is therefore important and justifies the need for research in this area.
The structure of the article is correct. The contribution was highlighted in the introduction. The research site and analysis results were well presented. The authors also indicated the practical application of the method and directions for further research. In my opinion, it can be published in its current version.
Kind regards
Author Response
Dear Reviewer,
Thank you very much for your positive comments and suggestions on our manuscript. We value your professional input and time, which has been very helpful in our research endeavours.
Your recognition and encouragement means a lot to us and contributes positively to our academic achievements and development. We greatly appreciate the valuable feedback and guidance you provided during the review process, which will help us improve and refine our research findings.
Thank you again for your patience and professionalism, and we are deeply grateful for your support and recognition of our work. We look forward to continuing to work with you to promote the development of academic research.
Best regards,
Sincerely yours
Ziqiang Zeng, Yupeng Sun, Xinru Zhang.